# Kisspeptin Stimulates the Pulsatile Secretion of Luteinizing Hormone (LH) during Postpartum Anestrus in Ewes Undergoing Continuous and Restricted Suckling

**DOI:** 10.3390/ani11092656

**Published:** 2021-09-09

**Authors:** José Manuel Hernández-Hernández, Graeme B. Martin, Carlos Miguel Becerril-Pérez, Arturo Pro-Martínez, César Cortez-Romero, Jaime Gallegos-Sánchez

**Affiliations:** 1Programa de Ganadería, Colegio de Postgraduados, Campus Montecillo, km 36.5 Carretera Federal México-Texcoco, Montecillo 56230, Mexico; josemanuel_hhc@yahoo.com.mx (J.M.H.-H.); aproma@colpos.mx (A.P.-M.); 2School of Agriculture and Environment, University of Western Australia, Crawley 6009, Australia; graeme.martin@uwa.edu.au; 3Colegio de Postgraduados, Campus Veracruz, Veracruz 91690, Mexico; color@colpos.mx; 4Colegio de Postgraduados, Campus San Luis Potosí, San Luís Potosi 78600, Mexico; ccortez@colpos.mx

**Keywords:** kisspeptin, LH, restricted suckling, anestrus, sheep

## Abstract

**Simple Summary:**

One of the major impediments to improving the efficiency of sheep production systems is the difficulty of breeding the females before their young are weaned. A major physiological barrier is suckling, because it prevents the initiation of a new reproductive cycle by inhibiting the pulsatile secretion of gonadotropin-releasing hormone (GnRH) and thus the secretion of LH pulses. It has now become clear that, at brain level, the secretion of GnRH is controlled primarily by the neuropeptide kisspeptin (Kp), a central player in the ‘KNDy’ system that generates GnRH pulses. Here, we report that intravenous administration of Kp stimulates pulsatile LH secretion in ewes during postpartum anestrus. Moreover, the response was greater when suckling was restricted to 30 min twice a day. We conclude that kisspeptin application increases pulsatile LH secretion in suckling ewes, suggesting that suckling inhibits ovulation by reducing the activity of kisspeptin neurons.

**Abstract:**

This study tested whether the intravenous application of kisspeptin can stimulate the pulsatile secretion of LH in suckling ewes during postpartum anestrus. Ten days after lambing, Pelibuey ewes were allocated among two groups: (1) continuous suckling (*n* = 8), where the lambs remained with their mothers; and (2) restricted suckling (*n* = 8), where the mothers suckled their lambs twice daily for 30 min. On Day 19 postpartum, the ewes were individually penned with *ad libitum* access to water and feed and given an indwelling catheter in each jugular vein. On Day 20, 4 mL of blood was sampled every 15 min from 08:00 to 20:00 h to determine LH pulse frequency. At 14:00 h, four ewes in each group received 120 μg of kisspeptin diluted in 3 mL of saline as a continuous infusion for 6 h; the remaining four ewes in each group received only saline. The interaction between kisspeptin and suckling type did not affect LH pulse frequency (*p* > 0.05). Before kisspeptin administration, pulse frequency was similar in all groups (1.50 ± 0.40 pulses per 6 h; *p* > 0.05). With the application of kisspeptin, pulse frequency increased to 3.50 ± 0.43 pulses per 6 h (*p* ≤ 0.014), so the concentration of LH (1.11 ± 0.14 ng mL^−1^) was greater in kisspeptin-treated ewes than in saline-treated ewes (0.724 ± 0.07 ng mL^−1^; *p* ≤ 0.040). The frequency of LH pulses was greater with restricted suckling than with continuous suckling (2.44 ± 0.29 versus 1.69 ± 0.29 pulses per 6 h; *p* ≤ 0.040). We conclude that intravenous application of kisspeptin increases the pulsatile secretion of LH in suckling ewes and that suckling might reduce kisspeptin neuronal activity, perhaps explaining the suppression of ovulation. Moreover, the effects of kisspeptin and suckling on pulsatile LH secretion appear to be independent, perhaps operating through different neural pathways.

## 1. Introduction

The duration of postpartum anestrus in sheep is influenced mainly by suckling [1], which inhibits the secretion of gonadotropin-releasing hormone (GnRH), thus reducing the frequency of pulses of GnRH and luteinizing hormone (LH) [2,3] but not affecting the secretion of FSH [4]. Restriction of suckling can reduce the duration of postpartum anestrus, increase the number of ovulating ewes and reduce the delay to first ovulation [5,6], with all of these responses implying an increase in the pulsatile secretion of GnRH/LH.

The inhibitory response to suckling begins with the lamb(s) sucking the mammary gland, evoking a stimulus that, along with visual and olfactory signals [7], travels through neural networks to reach the central nervous system (CNS). It is not known exactly which brain areas receive and respond to the inhibitory sucking stimulus in sheep but, from studies in rats, it seems likely that the preoptic area (POA) and the arcuate nucleus (ARC) are involved [8], as evidenced by the production of c-fos mRNA [9].

In addition to a poor understanding of the route taken by the sucking stimulus, it is not known if the pathway that inhibits pulsatile GnRH secretion in sheep is direct or passes via intermediary neuronal structures. On the other hand, it is known that estradiol (E_2_) inhibits the pulsatile secretion of GnRH/LH during lactational anestrus in sheep and seems to mediate the effect of suckling [10]. Critically, the effect of E_2_ on GnRH-producing neurons must be indirect, because those neurons do not have E_2_ type alpha receptors (ERα) [11]. This conundrum appears to have been resolved by the discovery that kisspeptin-neurokinin-dynorphin (KNDy) neurons provide E_2_ input into GnRH neurons [12].

In ewes, kisspeptin concentrations are high in the POA and the ARC, and kisspeptin neurons project towards the GnRH neurons of the POA, as well as the ARC and median eminence, where they regulate GnRH secretion [13,14,15]. Moreover, kisspeptin is involved in the regulation of GnRH/LH secretion during anovulatory states, such as seasonal anestrus and before puberty [16,17], when infusion with kisspeptin (20 µg h^−1^) is sufficient to increase LH pulse frequency and induce the preovulatory LH surge [18]. To date, this concept has not been extended to post-partum anovulation, although the KNDy system is likely to be involved because, in suckling rats, the expression of the KISS1 gene in the ARC is reduced and intraventricular application of kisspeptin increases LH pulse frequency and concentration [8].

Therefore, we tested whether intravenous kisspeptin would increase LH pulse frequency in suckling ewes.

## 2. Materials and Methods

### 2.1. Experiment Location

The experiment was carried out during August–September, the reproductive season for Pelibuey sheep, at the Laboratory of Sheep and Goat Reproduction (LaROCa) at Colegio de Postgraduados, Texcoco, Mexico State (98°53′ W, 19°29′ N; altitude 2240 m). The climate is temperate sub-humid with summer rains Cb(wo)(w)(i)g, average annual precipitation of 637 mm and average annual temperature of 15.2 °C [19].

### 2.2. Animals and Feeding

The study was conducted following the Official Mexican Standard (NOM-062-ZOO-1999) for technical specifications for the production, care and use of laboratory animals [20], as well as the regulations for the use and care of research animals issued by the General Academic Council of Colegio de Postgraduados in Mexico [21]. Sixteen multiparous Pelibuey ewes were used, 10 days after lambing, during which period they continuously suckled their young. Ewes were weighed at parturition (Day 0) and on Days 10, 17 and 24 postpartum. The weight of the ewes at lambing was 54.2 ± 1.5 kg. The ewes and lambs were housed indoors and fed with oaten hay (2 kg ewe^−1^ d^−1^) and feed concentrate (500 g ewe^−1^ d^−1^; Borrega Plus^®^, Unión Tepexpan, Tepexpan, Mexico) containing 15.2% crude protein (CP) and 2.5 Mcal of metabolizable energy (EM) kg^−1^. The lambs were suckled and offered free access to starter feed (Iniciador Dulce^®^, Nuevo León, Mexico).

### 2.3. Treatments

Four treatments (each *n* = 4) were used: (1) continuous suckling with saline infusion (CSS); (2) continuous suckling with kisspeptin infusion (CSK); (3) restricted suckling with saline infusion (RSS); (4) restricted suckling with kisspeptin infusion (RSK). On postpartum Day 10, ewes were randomly allocated to the continuous (C) and restricted (R) suckling groups. Ewes and lambs in the C groups remained in the same pen for 24 h per day. Ewes and lambs in the *R* groups were separated and, every day at 08:00 and 15:00 h, the ewes were taken to the pen containing the lambs for 30 min. On Day 19 postpartum, ewes within each of the C and R groups were randomly allocated to treatment with 3 mL saline or 120 µg kisspeptin in 3 mL saline [18]. For the infusions and serial blood sampling, catheters (1.1 mm internal diameter, BD Insyte, BD Vialon material, Franklin Lakes, NJ, USA) were inserted into both jugular veins of each ewe.

We used bovine kisspeptin-10 (YNWNSFGLRY) from Phoenix Pharmaceuticals Inc. (Burlingame, CA, USA) and, on Day 20 postpartum, infused it at a rate of 500 µL h^−1^ (20 µg h^−1^) [16] for 6 h beginning at 14:00 h. The treatments were administered with a constant infusion pump (NE-300, New Era Pump Systems, Farmingdale, NY, USA) connected to the right catheter by plastic tubing (1.2 m long and 2 mm internal diameter). At treatment termination, the ewes were housed in individual cages with access to water, food and space to lie down.

### 2.4. Blood Sampling

During the infusion, blood (4 mL) was sampled via the left catheter every 15 min from 08:00 h to 14:00 h (before infusion) and then from 14:00 h to 20:00 h (during infusion). The blood was placed in a tube without anticoagulant and immediately centrifuged at 700× *g* for 15 min (2500 rpm in a Solbat^®^ C-600 centrifuge, Solbat, Puebla, México). The resulting serum was decanted and stored at −20 °C until radioimmunoassay for LH.

### 2.5. Assay of LH

The serum concentration of LH was determined with the radioimmunoassay described by Arrieta [22], using ovine LH (oLH-1-2) from the National Institute of Diabetes and Digestive and Kidney Disease (Phoenix, AZ, USA) as a tracer. The limit of detection was 0.03 ng mL^−1^. The intra-assay and inter-assay coefficients of variation were 2.6 and 6.3%, respectively.

### 2.6. Statistical Analysis

The response variables were ewe live weight, LH pulse frequency, LH pulse amplitude and mean LH concentration. A pulse of LH was defined as a point which exceeded the previous measurement by two standard deviations (SD), with two consecutive high values being considered a single pulse. Pulse amplitude was determined as the difference between the LH value immediately before the onset of the pulse and the maximum value attained [23]. Ewe live weight data were analyzed using PROC MIXED (SAS 9.0, SAS Institute, Cary, NC, USA). The effects of kisspeptin and suckling treatments on LH secretion were analyzed as two periods (before and after), also using PROC MIXED. Means were compared using the Tukey test.

## 3. Results

### 3.1. Ewe Live Weights

Live weight was not affected by suckling type, infusion type or any interaction among treatments (*p* > 0.05). However, overall, the ewes lost weight (*p* ≤ 0.010) with each successive measurement as the experiment progressed: 55.9 ± 1.5 kg on Day 1; 53.7 ± 1.5 kg on Day 10; 51.8 ± 1.6 kg on Day 17; 50.3 ± 1.7 kg on Day 24.

### 3.2. LH Pulse Frequency

Examples of pulse profiles are shown in Figure 1, in which the robust response to kisspeptin infusion is clearly evident. There were no significant effects (*p* > 0.05) of the three-way interaction (suckling type × kisspeptin × sampling period) or the two-way interactions (kisspeptin × suckling type; suckling type × sampling period). However, the main effect of suckling was significant (*p* ≤ 0.026), with 0.8 more pulses per 6 h detected with restricted suckling than with continuous suckling (Table 1). The frequency of LH pulses was increased (*p* ≤ 0.006) by the infusion of kisspeptin (Table 2). There was a significant interaction between kisspeptin treatment and sampling period (*p* ≤ 0.007), with similar pulse frequencies observed before the start of infusion (*p* > 0.05), followed by higher frequencies during infusion with kisspeptin, as evidenced by comparison frequencies before and after the start of kisspeptin infusion (*p* ≤ 0.002), as well as comparison of frequencies during infusion of saline and kisspeptin (*p* ≤ 0.003).

### 3.3. LH Pulse Amplitude

Pulse amplitude was not significantly affected by any of the interactions among suckling treatment, infusion treatment, sampling period or by any of the main effects (Table 3).

### 3.4. Mean LH Concentration

During the infusion period, LH concentrations were greater in ewes that received kisspeptin than in those that received saline only (Table 4; *p* ≤ 0.040). The effect of the interaction between kisspeptin and period was not significant (*p* > 0.20). The effect of the interaction between kisspeptin and suckling type was not significant (*p* > 0.05) and there was no significant difference between restricted and continuous suckling (*p* > 0.05). The three-way interaction (kisspeptin × suckling type × period) was not significant (*p* > 0.05).

## 4. Discussion

Exogenous kisspeptin elicited a robust increase in LH pulse frequency in postpartum anestrus ewes, adding to the general consensus that this neuropeptide is a key stimulatory factor in the control of pulsatile GnRH secretion [16]. The frequency of LH pulses increased 1.3-fold more than in the control group. Moreover, this response was observed in more than half of the ewes in the kisspeptin group. The responses that we have observed during post-partum anovulation are similar in scale to those observed in other situations where ewes are anovulatory. For example, in prepubertal ewes, a similar dose of kisspeptin increased the LH pulse frequency from 0.5 to 1 pulses h^−1^ [17], as had been reported for suckling rats [8]. The increase in LH pulse frequency in the present study (from 0.25 to 0.58 pulses h^−1^) is smaller than in other studies. Although, the size of the response is physiologically relevant because it reflects a 130% greater stimulation of ovarian follicles. Moreover, in most of the sheep, there was a 230% increase in the gonadotrophic stimulus.

There is no logical explanation for the response to exogenous kisspeptin being smaller than expected, since the initial weight and the number of suckled young per ewe were similar. Overall, the ewes were in negative energy balance, a factor that could affect LH secretion [9]. Although, this factor does not fully explain the outcome, because the weights of the ewes were similar in all treatments. It is possible that, in some ewes, the 10% loss of live weight may be sufficient to inhibit the pulsatile secretion of GnRH/LH, reinforcing the suckling effect and reducing responses to kisspeptin. On the other hand, according to Lozano [24], a 12% loss of live weight does not decrease the frequency of LH pulses in postpartum anestrus ewes.

In the present study, the mean plasma concentration of LH was higher in ewes receiving kisspeptin than in the control ewes, as observed in seasonally anestrous ewes treated with a similar dose of kisspeptin [18]. The change in mean concentration probably reflects a change in pulse frequency, because pulse amplitude was not affected by kisspeptin application, in contrast with the responses of prepubertal ewes [17]. It has long been known that amplitude is difficult to measure with precision when a limited number of pulses are observed, because the measurement depends on sampling frequency [25], so the experimental protocol, as well as the physiological state of the animals and perhaps the LH standards used in the assays, could explain the differences between studies. In any case, GnRH/LH pulse amplitude does not change greatly throughout the estrous cycle [26], so it is probably not a major factor driving follicle development to ovulation [27], except in the release of preovulatory surge, as suggested by modelling [28] and demonstrated by detailed measurements of GnRH and LH [29].

Restriction of suckling did not influence the LH response to kisspeptin, but it did increase pulsatile LH secretion and, in a previous study, was even sufficient as a sole treatment to induce ovulation in postpartum ewes [5]. The similarity of the kisspeptin-induced increases in LH pulse frequency in the continuous-suckling and restricted-suckling groups suggests that the effects of kisspeptin and suckling are independent, perhaps operating through different neuroendocrine pathways. However, more work is needed to clarify the conditions needed for these responses. For example, Mandiki [4] detected no GnRH/LH response to the restriction of suckling, perhaps because, in their study, the ‘restriction’ was suckling three times per day, in contrast to the twice per day of the present study. In Pelibuey ewes, restriction of suckling to twice daily can increase the proportion of ewes ovulating from 35 to 57% during the first 60 d postpartum [1]. Reducing the suckling frequency to twice daily might be reducing the inhibitory effect of suckling on the hypothalamic expression of kisspeptin, as apparently occurs in rats [30].

## 5. Conclusions

Intravenous infusion of kisspeptin into Pelibuey ewes during postpartum anestrus increased the frequency of LH pulses, but the response was not affected by restriction of suckling to twice per day.

## Figures and Tables

**Figure 1 animals-11-02656-f001:**
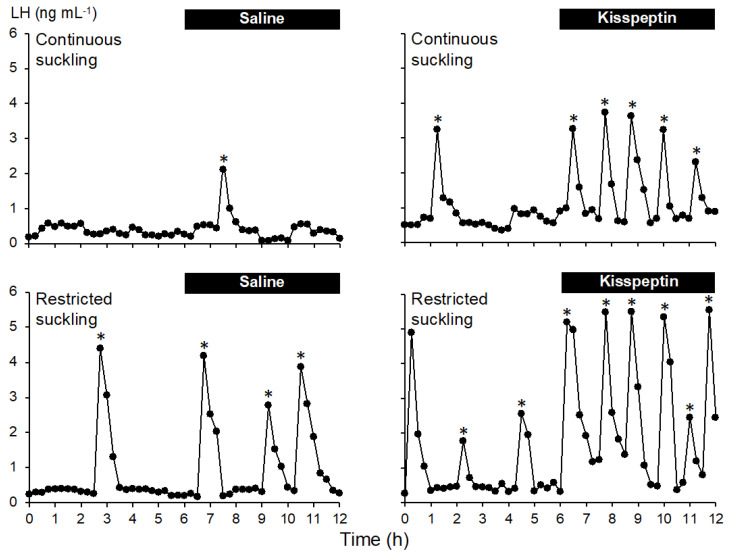
Examples of pulsatile LH secretion in postpartum anestrous ewes that were continuously suckling and infused with saline (**top left**), or kisspeptin (**top right**), or were suckling-restricted and infused with saline (**bottom left**) or kisspeptin (**bottom right**). The bars indicate the period of infusion. Asterisks indicate detected pulses.

**Table 1 animals-11-02656-t001:** Frequency of LH pulses in postpartum anestrus ewes in relation to suckling treatment and infusion with saline or kisspeptin (120 µg infused over 6 h). Pulse frequency was measured over 6 h, both before and during infusion treatment.

Suckling	*n*	Saline (Control)		Kisspeptin		Mean
		Before	During	Before	During	
Continuous	4	1.00 ± 0.41	1.25 ± 0.25	1.50 ± 0.29	3.50 ± 0.65	1.81 ± 0.32 ^a^
Restricted	4	2.25 ± 0.48	2.25 ± 0.25	1.75 ± 0.25	4.25 ± 0.75	2.63 ± 0.33 ^b^

Different letters (a, b) indicate statistically significant differences (*p* ≤ 0.05).

**Table 2 animals-11-02656-t002:** Frequency of LH pulses in postpartum anestrus ewes in relation to infusion with kisspeptin (120 µg over 6 h) or saline, with data pooled for suckling treatment. Pulse frequency was measured over 6 h, both before and during infusion.

	Before Infusion	After Infusion	Mean
**Saline (control)**	1.63 ± 0.38 ^ae^	1.75 ± 0.25 ^ae^	1.69 ± 0.22 ^e^
**Kisspeptin**	1.63 ± 0.18 ^ce^	3.88 ± 0.48 ^df^	2.75 ± 0.38 ^f^
**Mean**	1.63 ± 0.20 ^aa^	2.81 ± 0.38 ^b^	

By row (a, b; *p* ≤ 0.05, or c, d; *p* ≤ 0.01), or column (e, f; *p* ≤ 0.05), different letters indicate statistically significant differences.

**Table 3 animals-11-02656-t003:** Amplitude of LH pulses in postpartum anestrous ewes, in which suckling was continuous or restricted, before and during infusion of saline or kisspeptin.

Suckling	Saline Infusion (S)		Kisspeptin Infusion (Kp)		Mean
	Before	During	Before	During	
Continuous	3.54 ± 0.74	2.98 ± 0.42	2.57 ± 0.37	2.77 ± 0.46	2.96 ± 0.25
Restricted	2.69 ± 0.62	2.39 ± 0.57	3.37 ± 0.49	3.39 ± 0.47	2.95 ± 0.29
Mean	2.90 ± 0.30		3.02 ± 0.24		

S: 3 mL of saline solution for 6 h; Kp: 120 µg of Kp in 3 mL de S for 6 h; before: 6 h of sampling every 15 min before initiating treatment with S or Kp; after: 6 h of sampling every 15 min after initiating treatment with S or Kp.

**Table 4 animals-11-02656-t004:** Concentration of LH in postpartum anestrus ewes in relation to infusion with kisspeptin (120 µg over 6 h) or saline, with data pooled for suckling treatment. Concentration was measured over 6 h, both before and during infusion.

	Before Infusion	After Infusion	Mean
**Saline (Control)**	0.73 ± 0.11 ^a,e^	0.72 ± 0.10 ^a,e^	0.72 ± 0.07 ^e^
**Kisspeptin**	0.76 ± 0.05 ^c,e^	1.46 ± 0.21 ^d,f^	1.11 ± 0.14 ^f^
**Mean**	0.74 ± 0.06 ^a^	1.09 ± 0.15 ^b^	

By row (a, b; *p* ≤ 0.05, or c, d; *p* ≤ 0.01), or column (e, f; *p* ≤ 0.05), different letters indicate statistically significant differences.

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
