# Peer review of "Kisspeptin Stimulates the Pulsatile Secretion of Luteinizing Hormone (LH) during Postpartum Anestrus in Ewes Undergoing Continuous and Restricted Suckling"

_animals, 2021, doi:10.3390/ani11092656_

Round 1

Reviewer 1 Report

Dear authors, congratulations for your good manuscript, in general is a great work but I have some considerations for you: 

Line 15: Simple Summary is not fully.

Bibliographic references are not in correct mdpi format.

Line 189. Table 4. I recommend you that explain and clarify the content of the table.

Line 209 to 216. Coud you explain better your hypothesis about response to exogenous kisspeptin for?

Line 246: How you can support your conclusion about KNDy neuronal activity 246 in the ARC ?

Reviewer 2 Report

This is a manuscript where authors injected post-partum ewes in different sucking ways, with i.v. Kp-10 by slow-releasing pump and measured the responsivity of Kiss1r on GnRH neurons by LH pulsatility. The presentation of data is very confusing to this reviewer and, as a result, may not provide the best information to the authors (and potential readers) of this manuscript for arriving at conclusions. This reviewer's evaluation of the data lead to a different interpretation than that of the authors. In its present form, this manuscript is not suitable for publication.

In addition, authors use to extrapolate the data to neuroendocrinological suppositions. To do so molecular techniques are required (in situ hybridization, immunohistochemistry, and so on). The objective and hypothesis stated in the manuscript are different from the used experimental model. Major changes to the Introduction section are necessary in order to clearly state the motivation of these study and, more importantly, how this specific study adds to the knowledge body generated by previous studies.

 L15. I did not understand why authors did not provide the Simple Summary.

L29. How did the authors provide the Kp infusion continuously for six hours?

L30. Pattern if you want to work with centesimal or decimal p-value.

L38. “…and that suckling might reduce kisspeptin neuronal activity and therefore suppress ovulation. Moreover, the effects of kisspeptin and suckling on pulsatile LH secretion appear to be independent, perhaps operating through different neural pathways.” Do not extrapolate your conclusion. You did not measure Kisspeptin neurons circuitry at mediobasal-hypothalamus, follicle dynamics and ovulation.

L64. Double-check through the manuscript the use of words after you shorten them, for example E2 – Estradiol.

L75. Your experimental model does not provide the KNDy containing neuron role on GnRH/LH pulsatility in suckling ewes. Neurokinin by itself stimulates GnRH neuron. However, you just measure the responsivity of KISS1r on GnRH neuron to kisspeptin stimulus, during postpartum in different suckling systems.

L136. Re-do analysis using repeated measures on time.

Tables are very confuse for readers. In Table 1, create a column of the p-value for the factors, suckling type, kisspeptin treatment and suckling x treatment. Same for Table 2, 3 and 4.

Reviewer 3 Report

The study demonstrated the stimulating effect of exogenous kisspeptin on pulsatile release of LH from the pituitary into the peripheral blood in lactating sheep. A similar LH response profile to kisspeptin was recorded in sheep subjected to continuous suckling and in sheep with restricted suckling, where the mothers nursed their lambs twice daily.

The authors referred to the previously described effect of restrictive nursing of lambs on the length of postpartum anestrus in sheep. A novelty in this study is the emphasis on the role of kisspeptin, or rather its deficiency in early lactation, in the mechanism decreasing LH secretion.

In general, the mechanism responsible for increasing or delaying reproductive activity in sheep-mothers with different suckling regimes is not entirely clear. The authors note the similarity of the mechanisms responsible for postpartum anestrus and seasonal anestrus, including KNDy neurons and estradiol.

The presented work has many weaknesses and requires thorough changes. Without them, the value of the research will not merit publication in the journal ANIMALS.

  1. Introduction: the hypothesis is not adequate. The authors do not examine KNDy neurons, but only infuse kisspeptin intravenously; the authors overlook several other important mechanisms that may affect LH secretion during lactation, e.g. changes in the dopaminergic system induced by suckling – the involvement of salsolinol (Misztal et al. 2008, Marciniak et al. 2017);
  1. Material and methods: the number of animals per groups (n = 4) seems too small to clearly demonstrate the differences in the studied interactions (suckling type x kisspeptin x sampling period), (kisspeptin x suckling type; suckling type x sampling period); for the determination of LH pulses, the method for GnRH analysis was used, where samples are generally taken more frequently than 15 min. The Goodman et al. (2012) method is more adequate for LH; more details should be taken into account when describing the PROC MIXED analysis. How does it relate to the two/multivariate analysis of variance, whether parametric or non-parametric tests were used?
  1. Results: the significance level in the absence of differences can be omitted; line 185, P<0.20 is this significant?
  2. Discussion: It is difficult to conclude about the involvement of the KNDy system during lactation, when kisspeptin is silenced. The authors describe only the effects of administering exogenous kisspeptin; in addition, consideration should be given to whether kisspeptin passes from the blood to the brain? What does the term mean: “in more half of the ewes in the treatment group”? Line 200, more than two sheep? Also line 207 “in at least in half of the treated ewes” this indicates that the number of sheep is too small to state with full confidence about the result obtained; Consider involving other mechanisms (salsolinol), if not in the Introduction then in the Discussion.
  1. The conclusion is too extensive with additional speculation.It should be simple and concern the obtained results

Round 2

Reviewer 2 Report

I recognize authors' effort on this project, mainly blood sampling in this short interval is a method that is not easy to run. However, authors do not should to extrapolate discussion/conclusion for other stablished knowledge about Kp neuronal circuitry. As known by the scientific community, concluding based on the results obtained always increases the reliability of the article.

Reviewer 3 Report

For the clarity of the statistical method, mention could be made of the repeated measures analysis